# Nurse-Led Interventions in Chronic Obstructive Pulmonary Disease Patients: A Systematic Review and Meta-Analysis

**DOI:** 10.3390/ijerph19159101

**Published:** 2022-07-26

**Authors:** Alejandra Aranburu-Imatz, Juan de la Cruz López-Carrasco, Ana Moreno-Luque, José Manuel Jiménez-Pastor, María del Rocío Valverde-León, Francisco José Rodríguez-Cortés, Pedro Arévalo-Buitrago, Pablo Jesús López-Soto, Ignacio Morales-Cané

**Affiliations:** 1Department of Nursing, Maimonides Biomedical Research Institute of Cordoba (IMIBIC), University of Cordoba, 14004 Cordoba, Spain; ep2arima@uco.es (A.A.-I.); juanlopezcar7@gmail.com (J.d.l.C.L.-C.); n82molua@uco.es (A.M.-L.); jose_ma1991@hotmail.com (J.M.J.-P.); mariadelrociocordoba@gmail.com (M.d.R.V.-L.); francisco.rodriguez@imibic.org (F.J.R.-C.); z72arbup@uco.es (P.A.-B.); n82losop@uco.es (P.J.L.-S.); 2Department of Nursing, Pharmacology and Physiotherapy, University of Cordoba, 14004 Cordoba, Spain; 3Outpatient Clinic, Hospital Giovanni Paolo II, ULSS1 Dolomiti, 32044 Pieve di Cadore, Italy; 4Department of Nursing, Reina Sofia University Hospital, 14004 Cordoba, Spain

**Keywords:** chronic obstructive pulmonary disease, nurse interventions, nurse-collaborative interventions, nurse-led interventions, systematic review

## Abstract

Chronic obstructive pulmonary disease (COPD) is the third leading cause of death worldwide, causing 3.32 million deaths in 2019. COPD management has increasingly become a major component of general and hospital practice and has led to a different model of care. Nurse-led interventions have shown beneficial effects on COPD patient satisfaction and clinical outcomes. This systematic review was conducted to identify and assess nurse-led interventions in COPD patients in terms of mental, physical, and clinical status. The review was carried out following the Preferred Reporting Items for Systematic Reviews and Meta-analyses (PRISMA) statement. The relevance of each manuscript was assessed according to the inclusion criteria, and we retrieved full texts, as required, to reach our conclusions. Data extraction was performed independently by two reviewers, and the risk of bias was assessed using the Cochrane Risk of Bias tool. Forty-eight articles were included in the analysis, which focused on the management of COPD patients by hospital, respiratory and primary nursing care. Nursing management was shown to be highly effective in improving quality of life, emotional state, and pulmonary and physical capacity in COPD patients. In comparison, hospital and respiratory nurses carried out interventions with higher levels of effectiveness than community nurses.

## 1. Introduction

According to the World Health Organization [1], chronic diseases were the cause of 71% of deaths in the world in 2018 in ages ranging between 30 and 70 years old. Of these, chronic obstructive pulmonary disease (COPD) is the third leading cause of death worldwide, causing 3.23 million in 2019 [2].

Chronic obstructive pulmonary disease requires a complex, prolonged time period response, coordinating inputs from a wide range of professionals, specific kinds of drugs, and suitable monitoring equipment, and this care must ideally be embedded in a system that promotes patient empowerment. Many health systems are still largely built around an acute model of care, but the current challenge facing health policymakers is to figure out how to put in place a better response that meets COPD patient needs. As health systems differ widely, each must find its own solution. Even within similar systems, there may be marked differences in professional roles, coordination mechanisms, and care settings [3]. The national health systems have to face elderly multipathological population needs with a great economic and socio-health cost due to its great complexity. Multi-competence nurses accompany COPD patients in the management of all phases of the disease without generating an insurmountable economic burden for the coffers of the states, providing a good quality of care at home as well as in the hospital improving patient and family empowerment.

COPD management has increasingly become a key component of primary and hospital care leading to an evolving care model. International evidence has suggested that extending the role of primary and hospital nurses could improve the quality of life of COPD patients, adopting nurse-led care models to attain beneficial effects on patient satisfaction and clinical outcomes [4,5].

Fields of action in nursing range from health education to healthy lifestyles, correct management of inhalation pharmacology, early identification of signs of decompensation, respiratory rehabilitation, and palliative care. Nursing management of COPD patients achieves better outcomes in the day-to-day management of the disease, improving the patient’s knowledge of disease management [4,5]. Indeed, home visits and remote management with telemedicine at discharge time by hospital nurses and in primary care should be considered, not only for COPD patients but also for their families [6].

Given the benefits that nursing can bring to the care of COPD patients, it is essential to understand how nurse-led interventions affect COPD patients.

### Aim

The purpose of this review was to analyze the effect of hospital or community nurse-led interventions in the follow-up and management of COPD patients in terms of mental, physical, and clinical status.

## 2. Materials and Methods

### 2.1. Design

The systematic review and meta-analyses were carried out following the Preferred Reporting Items of Systematic Reviews and Meta-Analyses (PRISMA) statement [7]. The CENTRAL and PROSPERO databases were consulted to see if there were any systematic reviews with the same characteristics. The systematic review was complemented by a review of the references of the selected manuscripts. The meta-analysis protocol was registered in PROSPERO (CRD42020161153).

### 2.2. Search Strategy

A systematic search was carried out for manuscripts published between January 2009–January 2021 in PubMed, Embase, and Web of Science databases. The complete search strategy is detailed in Table 1.

### 2.3. Inclusion Criteria

The inclusion criteria were: (a) observational studies (case–control, cohort and cross-sectional) or intervention study (randomized or non-randomized); (b) manuscripts that analyzed patients who had COPD or diseases characterized by airflow limitation of a progressive and not easily reversible nature; (c) studies in which nursing-led intervention was carried out in both primary care and hospital care; (d) studies in which some health outcome was evaluated (quality of life, exacerbations, hospital admissions, emergency department admissions, patient empowerment in disease management, home management of terminal phase of disease, etc.). Articles that did not meet any of the above inclusion criteria were excluded.

### 2.4. Study Selection

Once the search strategies were defined, the search on the databases was carried out from 1 June 2021 to 30 September 2021. The evaluation period was decided in order to take into account COVID pandemic period, the effect it had had on these patients, therefore from 2009 to 2019 there were 10 years plus two years of pandemic. Two investigators (AAI and IMC) independently assessed all references identified in the search. Firstly, duplicates were eliminated, and screening was performed by reading titles and abstracts. Secondly, manuscripts that met the inclusion criteria in the first phase were read in full text to determine their inclusion. In case of discrepancies between the two investigators, a third party (PJLS) was consulted.

### 2.5. Data Collection

The information extracted from the studies included using a standardized form specifically designed for the purpose: author, title, year, description of patient population, sample size, duration of follow-up period, nurses’ involvement in study, characteristics of intervention, comparators, and outcomes.

The data were collected on a form created by the authors based on the PICO [8] strategy (population, intervention, comparator, results). The information extracted was: type of nurses (specialties) who work in primary or hospital settings, characteristics of interventions (health education, respiratory rehabilitation, telecare, palliative care assistance, and application of specific techniques), and psychophysical or clinical outcomes: hospital admissions, emergency department admissions, exacerbations and patient empowerment in disease management.

For the quantitative analysis, a meta-analysis was performed using the Mantel–Hansen effects method with a 95% confidence interval for dichotomous variables, while the inverse variance random effects method was used for continuous variables. Statistical heterogeneity was assessed with the I2 statistic. The cut-off of I^2^ ≤ 25%, I^2^ with 26–50%, and I^2^ > 50% was used to define low, moderate, and high heterogeneity, respectively [7]. Publication bias was assessed using funnel plots. Analyzes were performed with RevMan software, version 5.4.

### 2.6. Risk of Bias Assessment

The quality of the evidence was assessed using the Grading of Recommendations Assessment, Development and Evaluation (GRADE) [9] tool, classifying the quality of the evidence as low, moderate, or high, by evaluating the risk of bias, imprecision, directionality, inconsistency, and suspected publication bias.

The risk of bias was assessed using the Cochrane risk of bias assessment tool, which assesses selection bias, performance bias, attrition bias, reporting bias, and other biases. A funnel plot was made with the Cochrane Review Manager (Rev Man) 5.4 software from Cochrane. The operation was performed and reviewed by two members of the research group.

## 3. Results

### 3.1. Search Results

The study selection flowchart is shown in Figure 1. A total of 19,486 references were identified, of which 5518 were duplicates. After reading the title and abstracts of 13,968 references, 13,766 did not meet the inclusion criteria. Reading the full text of 202 manuscripts, 154 were discarded for the following reasons: (i) no access to the full text was possible; (ii) different study design; (iii) several pathologies were analyzed; (iv) the manuscripts were assessment studies, and (v) nursing did not lead the intervention. Finally, 48 studies met the inclusion criteria for qualitative analysis, of which 25 were considered for meta-analysis.

### 3.2. Characteristics of Studies Included

Studies were performed in 16 different countries from Europe, Asia, the Middle East, and Central and Northern America.

In the manuscripts included, the interventions were led by nurses in hospitals and in primary care settings by specialist respiratory nurses. The sample sizes of the studies ranged between 14 [10] to 516 participants [11] for a total of 5215 patients in 48 studies. The mean age of the intervention group was 60.85 years old, while in the control group it was 54.18 years old. In general terms, the populations included in the studies were homogeneous.

The participants were recruited on hospital discharge following an exacerbation, or from community hospitals. Inclusion criteria were based on a clinical diagnosis of COPD, and most studies also required there to be an airflow obstruction confirmed by spirometry. The minimum age for inclusion in studies varied from 18 to 80 years old. Sex distribution was variable across the studies (5 to 94% of males). Follow-up length also varied among studies, ranging from 8 weeks to 3 years.

#### 3.2.1. Interventions

The review shows an upward trend in the number of publications since 2015. Heterogeneity was observed as regards the type of interventions and scope of care. The characteristics of the 48 studies selected are shown in Appendix A.

Specifically, twenty studies [11,12,13,14,15,16,17,18,19,20,21,22,23,24,25,26,27,28,29,30] reported data on hospital nurse management with a specific care plan of pharmacological management to empower COPD patients, i.e., correctly performed inhalation therapy training, use of oxygen therapy, and physical exercise, which are typically given on post-discharge home visits.

Twelve studies [11,19,30,31,32,33,34,35,36,37,38,39] analyzed the management of primary care nurses, the empowerment of COPD patients in activities of daily living, telemedicine, follow-up, and palliative care. Telemedicine for vital parameter monitoring and teleconsultation was led by hospital and community nurses in six manuscripts [40,41,42,43,44,45]. Additionally, the effect of nurse-led physical and respiratory rehabilitation interventions was analyzed in five studies [27,46,47,48,49].

Two studies [50,51] analyzed the collaboration between primary care nurses and hospital nurses at discharge time and the follow-up of COPD patients in the management of COPD disease, and four studies [15,16,52,53] analyzed inhalation drug training techniques given by specialized nurses with COPD patients.

#### 3.2.2. Outcomes

The outcomes analyzed by the selected studies varied widely and focused on the psychophysical and clinical status of people with COPD: lung and vital capacity, dyspnea, mortality, hospital admission, use of health services, compliance and knowledge of drug therapy, use of oxygen therapy, patient satisfaction, health literacy, physical status, quality of life, anxiety, and depression. A quantitative analysis (meta-analysis) was performed on four specific outcomes: hospital admission, physical status, quality of life, and anxiety.

##### Hospital Admissions

The number of hospital admissions after the intervention was analyzed in seven studies [11,15,29,41,42,43,54] with 959 patients, after 3 months, 6 months, 12 months, and 18 months: the hospitalizations tended to be lower in the intervention group than in the control group [standard mean difference −0.44 (95% CI −0.92, 0.04), *p* = 0.07; I^2^ = 87%], although heterogeneity was high (Figure 2).

##### Physical Status

The 6 m walking test (6 MWT) was used to verify the improvement of physical resistance in five studies [27,29,46,47,49] with 462 patients, after 3, 6 and 12 months: the patients improved in distance walked [SMD 0.66 (95% CI 0.10, 1.22), *p* = 0.02; I^2^ = 82%] (Figure 2).

Indeed, the Moderate to Vigorous Physical Activity (MVPA) and the International Physical Activity Questionnaire (IPAQ) were used in one study [11] with 411 patients, where after 12 months patients improved their physical activity, although the result was not statistically significant in the (MVPA) test (*p* = 0.48) and (IPAQ) test (*p* = 0.21). A walking diary was used in one study [47] with 24 patients, improving the number of minutes of walking per day at 8 weeks.

##### Quality of Life

Different tests were used to analyze the patient’s quality of life in 32 studies [10,11,12,13,16,19,21,22,23,24,25,26,27,28,29,30,31,32,33,34,35,37,38,40,46,47,48,50,51,52,54,55], such as the St. George’s Respiratory Questionnaire, Health Belief Scale, Self-Efficacy Scale, Stanford Self-Efficacy Score or World Health Organization Quality of Life Scale Abbreviated Version (QoL), among others.

Regarding St. George’s Respiratory Questionnaire, eleven studies [11,16,19,22,23,29,46,50,51,55,56] used it to assess quality of life (1.423 patients). Overall, no improvements were found in the intervention group when using this scale [SMD −0.11 (95% CI −0.95, 0.73); *p* = 0.79; I^2^ = 90%]. However, studies which used the Barthel scale to assess the impact on activities of daily living [27,48] showed an improvement after nurse-led intervention [SMD 1.36 (95% CI 0.87, 1.74); *p* < 0.01; I^2^ = 0%] (Figure 3).

##### Anxiety

Anxiety was analyzed using different scales in eight studies [10,11,17,19,28,30,37,54], such as the hospital anxiety and depression scale (HADS), anxiety and depression symptoms inventory scale (SCL), self-rating anxiety scale (SAS), and hospital anxiety scale (HAC).

Hospital anxiety and depression scale (HADS) was used in five studies [10,11,14,17,37] with 746 patients. Anxiety levels decreased in all the studies after the nurse-led intervention, except in Scheerens et al. [37]. Overall, the decrease in anxiety levels was significant [SMD −0.21 (95% CI −0.35, −0.07); *p* = 0.003; I^2^ = 0%] (Figure 3).

##### Risk of Bias in Selected Studies

Almost all the studies (*n* = 48) have a moderate risk of bias due to the study design. Eighteen manuscripts [10,13,14,15,16,25,28,30,31,32,39,41,46,50,52,53,54,55] were not randomized and five did not have a control group [14,25,31,52,53].

Due to the characteristics of the studies where the nurse had to follow and train patients, the double-blind was impossible; however, performance bias was accepted as correct or not present in twelve studies and there was a selection bias in 10% of studies [13,36,45,50,54]. Overall, five studies were considered as high methodological quality [21,22,29,33,34,40]. A detection bias was present in 75% studies [10,12,13,14,15,16,17,18,19,20,23,24,26,27,28,30,32,36,37,38,39,41,42,43,44,45,46,47,48,49,50,51,56,57].

## 4. Discussion

This systematic review and meta-analysis analyzed different types of nurse-led interventions in COPD patients which obtained improvements in physical status, quality of life, and anxiety and/or reductions in hospital admissions. Although there are several systematic reviews on the effects of interventions in the management of COPD patients, to our knowledge this is the first systematic review to analyze the effect of nurse-led interventions in this population.

As mentioned, interventions are led by different types of nurses: general hospital, community, palliative, and respiratory nurses [58]. In general, these professionals work in different areas using different techniques: home telemonitoring, telecare, palliative care both in hospital and at home, health education through continuous training or training in inhalation techniques and oxygen management at the home, hospital and home respiratory rehabilitation or telerehabilitation, self-efficacy and training in smoking cessation techniques.

Hospital nurse-led interventions use various types of nursing care models, including the health belief model; humanistic nursing care model; self-management education model; bidirectional quality feedback nursing model; psychological, cognitive-behavioral model; physical-functional model, and a nursing care model based on the information, knowledge, attitude, and practice (IKAP) theory. In general, the studies analyzed reports that these models had improved patients’ health status by empowering them to manage the disease.

Early identification of signs of COPD decompensation is a fundamental part of patient education, as is training on the correct use of pharmacological therapy, especially in inhalation techniques such as the use of oxygen therapy. In the present review, all the nurse-led interventions have been shown to have a positive impact on the patients’ stress levels and the number of hospitalizations [12,13,15,20,21,22,24,27,28,29,50,51,55,57].

Incorrect inhaler technique is very common among COPD patients, resulting in decreased efficacy of drug therapy and worsening patient health [59]. Inhaler training by the specialized nurse has been shown to significantly improve patient performance [16,53].

Anxiety and depression are common symptoms among COPD patients. The uncertainty of not knowing when an acute breathlessness attack may occur worsens the patient’s emotional situation, leaving them to cope with the anxiety and depression it produces. These findings suggest that psychological therapy, including cognitive behavioral therapy and counseling, can improve depressive and anxiety symptoms in COPD patients [60,61]. The present review also reports that nurses using special techniques such as cognitive behavioral therapy or minimal psychological intervention help the patients learn to manage anxiety successfully [11,14,17,19,28,30,55].

Home visits by the hospital nurse after hospital discharge have not shown a significant decrease in the number of readmissions, but patients reported feeling better and more confident, with greater knowledge of disease management and decreasing anxiety levels [12,50,52,55].

Such monitoring and telemonitoring of vital parameters and teleconsultation by the hospital nurse has reduced the number of admissions, days of hospitalization, access to emergency services, reduced levels of anxiety and depression, and improved quality of life [11,25,33,42,43,44,45,56]. This intervention, for a limited period of fewer than 6 months, has demonstrated efficacy. The only study that required more time was that by Taylor et al. [62], which increased morale but not the patient quality of life after one year of follow-up.

Community nurse-led education and telecare only resulted in reducing the number of days of hospitalization [41] and level of dyspnea—in the Medical Research Council Questionnaire and SF-36 physical score fields [30]—while the FEV1%, FEV1 FVC% ratio, FEV1 (L), number of ED visits were worse compared to the control group in several studies [38,40]. Similarly, the study by Baker et al. [63] in a primary care setting shows benefits in the management of COPD patients, decreased number of physician visits and anxiety, but the results were not significant and therefore no firm conclusions on efficacy can be drawn.

Respiratory rehabilitation in home telecare with different types of tools, such as mobile phones and the Bandura technique, improved patient quality of life in the study [46], in which the FEV1 FC1% ratio, baseline dyspnea index (BDI), and 6MWT scale scores improved; in the studies by Nguyen et al. [23], Khoshkesht et al. [36] and Mohammadi et al. [48], the self-efficacy scale, Barthel scale, fatigue severity scale (FSS) and SF36 scale also improved; only the study by Cameron-Tuker et al. [47] showed no improvement in smoking, nutrition, alcohol consumption, physical activity, psychosocial well-being or the symptom management scales.

Last but not least, evidence of the benefits of smoking cessation in COPD has been demonstrated, including decreased disease progression, and reduced symptoms and mortality [6]. Nurse interventions focused on reducing smoking have been covered in two studies [11,49] but only in Jolly et al. [11], the number of smokers decreased after 12 months.

This systematic review has several limitations. Although the search strategy has been exhaustive, using several keywords and three databases, it may be possible that not all the articles related to the study topic have been included. On the other hand, although the number of articles included is large, the heterogeneity in the study designs, the different outcome variables, and the different measurement tools have made it difficult to carry out a quantitative analysis. Finally, the evaluation of the quality of the evidence reported a medium level. Therefore, the findings obtained in this review must be taken with caution.

## 5. Conclusions

Nurse-led interventions in different fields, using different techniques and approaches, have demonstrated their effectiveness in improving quality of life, emotional state, hospital admissions, and physical capacity in COPD patients. In comparison, hospital nursing and respiratory nurse management have presented levels of effectiveness compared to community ones. However, these conclusions may be affected by the heterogeneity of the studies and their level of evidence, so a greater number of quality studies addressing this issue would be necessary.

Regarding the implications for clinical practice, this manuscript provides a broad view of the interventions that nurses could carry out in both community and hospital care.

The findings invite us to motivate, assess, and guide future nursing professionals to design steps to control and monitor chronic pathologies. Apparently simple actions such as health education can provide great benefits for both the patient and their family.

## Figures and Tables

**Figure 1 ijerph-19-09101-f001:**
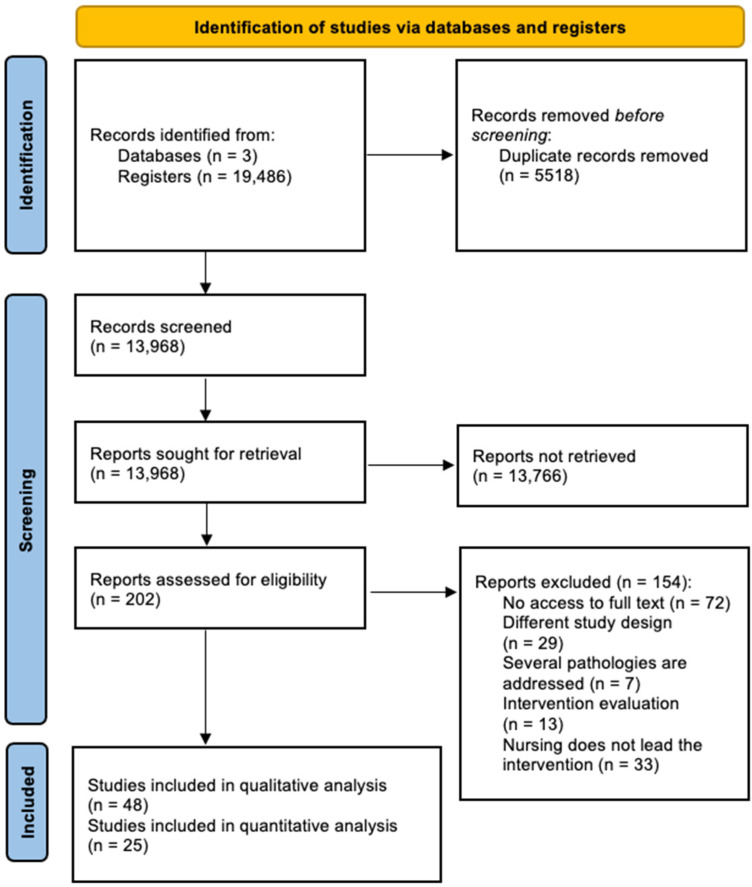
PRISMA flowchart of the study selection process.

**Figure 2 ijerph-19-09101-f002:**
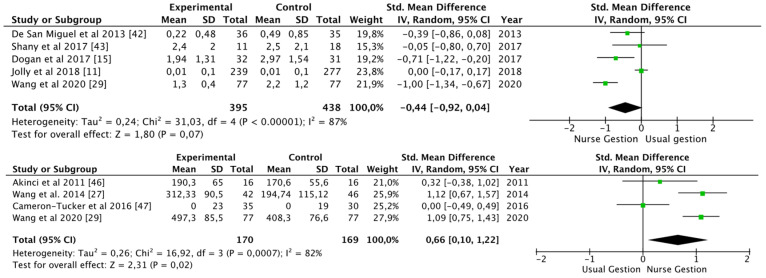
Number of hospital admissions and distance walked (6 MWT) [11,15,27,29,42,43,46,47].

**Figure 3 ijerph-19-09101-f003:**
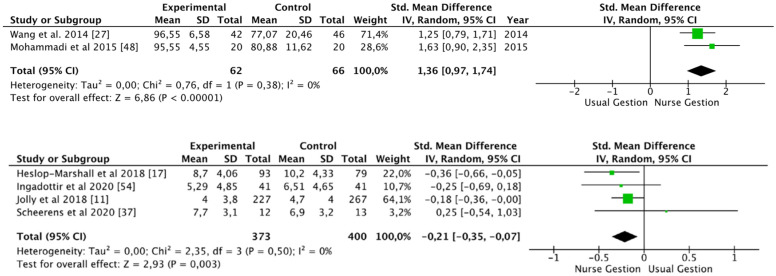
Activities of daily living (Barthel scale) and hospital anxiety and depression scale [11,17,27,37,48,54].

**Table 1 ijerph-19-09101-t001:** Search strategy used for each database.

Database	Search Strategy
PubMed	(“community nursing” [tiab] OR “family nursing” [tiab] OR “respiratory nursing” [tiab] OR “community nurse” [tiab] OR “family nurse” [tiab] OR “respiratory nurse” [tiab] OR “community nurses” [tiab] OR “family nurses” [tiab] OR “respiratory nurses” [tiab] OR “nurse” [tiab] OR “nursing” [Mesh] OR “nursing” [tiab] OR “nurses” [tiab]) AND (“copd” [tiab] OR “asthma” [tiab] OR “chronic bronchitis” [tiab] OR “chronic bronchitises” [tiab] OR “pulmonary emphysema” [tiab] OR “pulmonary emphysema” [tiab] OR “focal emphysema” [tiab] OR “focal emphysema” [tiab] OR “ panacinar emphysema” [tiab] OR “panacinar emphysema” [tiab] OR “panlobular emphysema” [tiab] OR “panlobular emphysema” [tiab] OR “centriacinar emphysema” [tiab] OR “centriacinar emphysema” [tiab] OR “centrilobular emphysema” [tiab] OR “centrilobular emphysemas” [tiab] OR “chronic obstructive pulmonary disease” [tiab] OR “chronic obstructive pulmonary” [tiab] OR “pulmonary disease, chronic obstructive” [Mesh])
EMBASE	“community nursing”:ti,ab OR “family nursing”:ti,ab OR “respiratory nursing”:ti,ab OR “community nurse”:ti,ab OR “family nurse”:ti,ab OR “respiratory nurse”:ti,ab OR “community nurses”:ti,ab OR “family nurse”:ti,ab OR “respiratory nurses”:ti,ab OR “nurse”:ti,ab OR “nursing”:ti,ab OR “nurses”) AND (“copd”:ti,ab OR “asthma”:ti,ab OR “chronic bronchitis”:ti,ab OR “chronic bronchitises”:ti,ab OR “emphysema”:ti,ab OR “emphysemas”:ti,ab OR “chronic obstructive pulmonary disease”:ti,ab OR “chronic obstructive pulmonary diseases”)
Web of Science	(AB = (community nursing) OR AB = (family nursing) OR AB = (respiratory nursing) OR AB = (community nurse) OR AB = (family nurse) OR AB = (respiratory nurse) OR AB = (community nurses) OR AB = (family nurses) OR AB = (respiratory nurses) OR AB = (nurse) OR AB = (nursing) OR AB = (nurses)) AND (AB = (copd” OR AB = (asthma” OR AB = (chronic bronchitis” OR (AB = (chronic bronchitises) OR AB = (pulmonary emphysema) OR AB = (pulmonary emphysema) OR AB = (focal emphysema) OR AB = (focal emphysema) OR AB = (panacinar emphysema) OR AB = (panacinar emphysema) OR AB = (panlobular emphysema) OR AB = (anlobular emphysema) OR AB = (centriacinar emphysema) OR AB = (centriacinar emphysema) OR AB = (centrilobular emphysema) OR AB = (centrilobular emphysemas) OR AB = (chronic obstructive pulmonary disease) OR AB = (chronic obstructive pulmonary))

## Data Availability

Not applicable.

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
