# Peer review of "Nurse-Led Interventions in Chronic Obstructive Pulmonary Disease Patients: A Systematic Review and Meta-Analysis"

_ijerph, 2022, doi:10.3390/ijerph19159101_

Round 1
Reviewer 1 Report
.- Topic of great interest for possible readers of the Journal. COPD is a highly prevalent, complex and heterogeneous disease with a high impact on patients and also on public health systems. The role of nursing in the follow-up and management of this pulmonar disease is essential. Congratulations on the submitted manuscript!
.- Some minor comments are made in favor of improving the current version of the manuscript:
.-Abstract. When he says “Nurse-led interventions have had…”. Evaluate change the phrase. “have had” (cacophony)
.- Keywords. Second word “Nurse intervention”. Evaluate changing for “nurse interventions”.
.- Methodology. A question arises, why did you choose the 2009-2021 as a period of study and not another, e.g. 2005-2021?
.- Figures. The manuscript presents 6 figures. Figures 2-5, maybe group them two by two ones. Consider increasing its size because it is very difficult to read all the fields clearly. Figure 6. Perhaps it could be removed. The biases of the studies are well explained in the previous paragraph.
.- Discussion. Perhaps, the second part (page 9 of 13) is long. Evaluate delimiting, above all, sections on telemonitoring and pulmonary rehabilitation.
.- Conclusion. Perhaps make a succinct mention of the limitations of the studies: heterogeneity of the studies, grade of evidence and interpret the results of the review with caution.
.- Supplementary material. Can it be interpreted that the link provided is not definitive? “www.mdpi.com/xxx/s1”. Table S1, its meaning is not final either, right?
.- References. The number of quotes provided is appreciated. However, of the 68 citations, only 19 are recent (year of publication equal to or less than 5 years). To consider, add some recent additional citation.
Last dates of access in the case of the references 1-3, better homogenize.
Review all references that are presented in compliance with the regulations of this journal. For example references 13, 53 or 56, the year of publication appears after the authors, while in the rest of the citations it appears at the end.
Reference 38. Indicates doi, Epub and PMID. In the rest of the citations these references do not appear. Evaluate remove.
Reference 67. Evaluate limiting/reducing the number of authors.
Author Response
.- Topic of great interest for possible readers of the Journal. COPD is a highly prevalent, complex and heterogeneous disease with a high impact on patients and also on public health systems. The role of nursing in the follow-up and management of this pulmonar disease is essential. Congratulations on the submitted manuscript!
Thank you very much for your positive comments.
.- Some minor comments are made in favor of improving the current version of the manuscript:
.-Abstract. When he says “Nurse-led interventions have had…”. Evaluate change the phrase. “have had” (cacophony)
We modified for “have shown”
.- Keywords. Second word “Nurse intervention”. Evaluate changing for “nurse interventions”.
We changed for “nurse interventions”
.- Methodology. A question arises, why did you choose the 2009-2021 as a period of study and not another, e.g. 2005-2021?
Reason was provided in methodology section
.- Figures. The manuscript presents 6 figures. Figures 2-5, maybe group them two by two ones. Consider increasing its size because it is very difficult to read all the fields clearly. Figure 6. Perhaps it could be removed. The biases of the studies are well explained in the previous paragraph.
We grouped Figure 2 and 3, and 4 and 5. Figure 6 was removed.
.- Discussion. Perhaps, the second part (page 9 of 13) is long. Evaluate delimiting, above all, sections on telemonitoring and pulmonary rehabilitation.
We removed sections on telemonitoring and pulmonary rehabilitation.
.- Conclusion. Perhaps make a succinct mention of the limitations of the studies: heterogeneity of the studies, grade of evidence and interpret the results of the review with caution.
A paragraph make a succinct mention on the study limitations.
.- Supplementary material. Can it be interpreted that the link provided is not definitive? “www.mdpi.com/xxx/s1”. Table S1, its meaning is not final either, right?
Yes, it is not final.
.- References. The number of quotes provided is appreciated. However, of the 68 citations, only 19 are recent (year of publication equal to or less than 5 years). To consider, add some recent additional citation.
Last dates of access in the case of the references 1-3, better homogenize.
Review all references that are presented in compliance with the regulations of this journal. For example references 13, 53 or 56, the year of publication appears after the authors, while in the rest of the citations it appears at the end.
Reference 38. Indicates doi, Epub and PMID. In the rest of the citations these references do not appear. Evaluate remove.
Reference 67. Evaluate limiting/reducing the number of authors.
References were updated and modified; also new ones were added.
Reviewer 2 Report
Thank you for the opportunity to read this manuscript and congratulations to the authors for their work. I highly appreciate the text submitted for review.
Here are some suggestions for improvement:
This reviewer suggests deleting "of this systematic, meta-analysis review" in line 65 (section 1.1. Aim), to include it in the Materials and Methods section (2.1. Design).
In Figure 1. PRISMA flowchart (page 4 of 13), please review the "screening section". Maybe records screened and reports sought for retrieval might be summed up (n = 13,968). On the other hand, 13,766 reports not retrieved might be better explained, and also, no access to full text (n =72) might not be a good reason to exclude some reports. The whole reports excluded are 154, but in line 142 (Results section, page 5 of 13), it is stated that 153 were discarded.
Some inexplicable style mistakes are observed. Examples are given in the last two paragraphs of page 6 of 13: uses of acronyms (MVPA or IPAQ), or the numbers of references (lines 217-218) among others. Review and correct intratext references (e.g.: "[8, -11, 14, 17, 19-33, 35, 36, 38, 44, 45, 46, 48, 49, 50]".
In lines 352-353 (page 10 fo 13) it is stated that "Supplementary Materials: The following supporting information can be downloaded at: www.mdpi.com/xxx/s1, Table S1: Characteristics of studies focusing on nurse-led COPD patients; Figure S1. Assessment of publication bias. Funnel Plots; Figure S2. Risk of bias of all selected studies". Please, provide the aforementioned supplementary materials. Last but not least, this reviewer would suggest including here the summary of findings (SoF) tables.
Author Response
Reviewer 2
Thank you for the opportunity to read this manuscript and congratulations to the authors for their work. I highly appreciate the text submitted for review.
Thank you very much for your positive comments.
Here are some suggestions for improvement:
This reviewer suggests deleting "of this systematic, meta-analysis review" in line 65 (section 1.1. Aim), to include it in the Materials and Methods section (2.1. Design).
Modifications were added.
In Figure 1. PRISMA flowchart (page 4 of 13), please review the "screening section". Maybe records screened and reports sought for retrieval might be summed up (n = 13,968). On the other hand, 13,766 reports not retrieved might be better explained, and also, no access to full text (n =72) might not be a good reason to exclude some reports. The whole reports excluded are 154, but in line 142 (Results section, page 5 of 13), it is stated that 153 were discarded.
We followed the Cochrane guidelines.153 was a typo. We added the proper number (154)
Some inexplicable style mistakes are observed. Examples are given in the last two paragraphs of page 6 of 13: uses of acronyms (MVPA or IPAQ), or the numbers of references (lines 217-218) among others. Review and correct intratext references (e.g.: "[8, -11, 14, 17, 19-33, 35, 36, 38, 44, 45, 46, 48, 49, 50]".
The changes in numbering were made.
In lines 352-353 (page 10 fo 13) it is stated that "Supplementary Materials: The following supporting information can be downloaded at: www.mdpi.com/xxx/s1, Table S1: Characteristics of studies focusing on nurse-led COPD patients; Figure S1. Assessment of publication bias. Funnel Plots; Figure S2. Risk of bias of all selected studies". Please, provide the aforementioned supplementary materials. Last but not least, this reviewer would suggest including here the summary of findings (SoF) tables.
At the editor's request, it was provided by email. We apologise for the inconvenience.
Reviewer 3 Report
This manuscript presents the results of a systematic review and meta-analysis purporting to analyze nurse-led interventions in COPD patients in terms of mental, physical, and clinical status. The topic is relevant and the study design and procedure are very clear. The article has clear language and the aim of the study is clear and interesting. I have only minor suggestions for revision:
In the title, I recommend that the authors spell out COPD for those readers who do not know this abbreviation.
Introduction
The introduction needs to be improved, with background information on the importance of this intervention for COPD patients and health system performance. What is the added health benefit for the patient? What are the positive effects on patients? These are some questions that need to be answered in the introduction.
Material and methods
Line 70- “The review was carried out following the Preferred Reporting Items of Systematic 70 Reviews and Meta-Analyses (PRISMA) statement”- Please provide a reference at the end of the sentence.
Line 78- “A systematic search was carried out for manuscripts published between January 2009 78 -January 2021 in PubMed, Embase and Web of Science databases.”- why the authors chose this temporal context? This aspect needs to be clarified.
Figure 1- When identifying articles, authors must indicate the number of articles identified by each database used for the research.
Line 112- “The data was collected on a form created by the authors based on the PICO strategy (Population, Intervention, Comparator, Results).” Please provide a reference at the end of the sentence.
Line 128- “The quality of the evidence was assessed using the Grading of Recommendations 128 Assessment, Development and Evaluation (GRADE) tool, classifying the quality of the evidence as low, moderate, or high, by evaluating the risk of bias, imprecision, directionality, inconsistency, and suspected publication bias”. Please provide a reference at the end of the sentence.
Author Response
Reviewer 3
This manuscript presents the results of a systematic review and meta-analysis purporting to analyze nurse-led interventions in COPD patients in terms of mental, physical, and clinical status. The topic is relevant and the study design and procedure are very clear. The article has clear language and the aim of the study is clear and interesting. I have only minor suggestions for revision:
Thank you very much for your positive comments.
In the title, I recommend that the authors spell out COPD for those readers who do not know this abbreviation.
We modified the title.
Introduction
The introduction needs to be improved, with background information on the importance of this intervention for COPD patients and health system performance. What is the added health benefit for the patient? What are the positive effects on patients? These are some questions that need to be answered in the introduction.
The economic and effective management motivations on the part of the nurse were added.
Material and methods
Line 70- “The review was carried out following the Preferred Reporting Items of Systematic 70 Reviews and Meta-Analyses (PRISMA) statement”- Please provide a reference at the end of the sentence.
We provided such reference.
Line 78- “A systematic search was carried out for manuscripts published between January 2009 78 -January 2021 in PubMed, Embase and Web of Science databases.”- why the authors chose this temporal context? This aspect needs to be clarified.
As referred to reviewer 1, the reason for the number of years has been added in the methodology section.
Figure 1- When identifying articles, authors must indicate the number of articles identified by each database used for the research.
We followed the Cochrane and PRISMA guidelines.
Line 112- “The data was collected on a form created by the authors based on the PICO strategy (Population, Intervention, Comparator, Results).” Please provide a reference at the end of the sentence.
Reference was added.
Line 128- “The quality of the evidence was assessed using the Grading of Recommendations 128 Assessment, Development and Evaluation (GRADE) tool, classifying the quality of the evidence as low, moderate, or high, by evaluating the risk of bias, imprecision, directionality, inconsistency, and suspected publication bias”. Please provide a reference at the end of the sentence.
References was added.
Reviewer 4 Report
Dear Editor and Authors,
Thank you for asking me to review this manuscript titled “Nurse-led interventions in COPD patients: a systematic review and meta-analysis” by Alejandra Aranburu-Imatz and her colleagues from the Department of Nursing at the Maimonides Biomedical Research Institute of Cordoba (IMIBIC) in Córdoba, Spain.
In this work the authors conduct a review and a meta-analysis of the literature to identify and assess nurse-led interventions in chronic obstructive pulmonary disease (COPD) patients using the widely accepted PRISMA guidelines.
This is a well conducted meta-analysis with a robust as described methodology following accepted protocol. The authors have sought to analyze a number of clinical outcomes utilizing patients from each study. One question which comes to mind is if the authors were able to assert if the number of patients (sample size) for each outcome was adequate to provide statistically meaningful results – in sort an individual power analysis for each outcome. This would strengthen as you can realize the validity of the report.
Considering the wide variety of nurse lead interventions reported in the studies it is obvious that this heterogeneity of actions create a significant bias – the authors have well recognized this but I was wondering if a model could be constructed to assess the effect of it!
Similarly the wide variety of test utilized to access quality of life make difficult the analysis and comparison of outcomes. This is again a significant limitation which needs addressing.
Overall, this is a well written manuscript, clear and concise. It attempts to deal with a very wide subject range and has significant bias and heterogeneity in both outcomes and measuring tools – this is of course expected given that this is a meta-analysis of different studies. The authors have addressed this in their limitation section but I would be more happy if they elaborated a bit more about it in their discussion. I do feel this is an interesting study and it merits publication provided some minor corrections as mentioned are made!! It is not a groundbreaking study and not the most robust analysis – not because correct methodology was not followed but due to the nature of the available data and studies. Nevertheless I have not seen something similar come out and therefore I think some evidence is better than no evidence – provided limitations are addressed/mentioned for the reader to be aware.
In conclusion, good job to the authors, I am awaiting your revised work. Try to see if sub-power analysis can be performed – it will add to the work! Kind regards and stay healthy.
Author Response
Reviewer 4
Dear Editor and Authors,
Thank you for asking me to review this manuscript titled “Nurse-led interventions in COPD patients: a systematic review and meta-analysis” by Alejandra Aranburu-Imatz and her colleagues from the Department of Nursing at the Maimonides Biomedical Research Institute of Cordoba (IMIBIC) in Córdoba, Spain.
In this work the authors conduct a review and a meta-analysis of the literature to identify and assess nurse-led interventions in chronic obstructive pulmonary disease (COPD) patients using the widely accepted PRISMA guidelines.
This is a well conducted meta-analysis with a robust as described methodology following accepted protocol. The authors have sought to analyze a number of clinical outcomes utilizing patients from each study. One question which comes to mind is if the authors were able to assert if the number of patients (sample size) for each outcome was adequate to provide statistically meaningful results – in sort an individual power analysis for each outcome. This would strengthen as you can realize the validity of the report.
Considering the wide variety of nurse lead interventions reported in the studies it is obvious that this heterogeneity of actions create a significant bias – the authors have well recognized this but I was wondering if a model could be constructed to assess the effect of it!
Similarly the wide variety of test utilized to access quality of life make difficult the analysis and comparison of outcomes. This is again a significant limitation which needs addressing.
Overall, this is a well written manuscript, clear and concise. It attempts to deal with a very wide subject range and has significant bias and heterogeneity in both outcomes and measuring tools – this is of course expected given that this is a meta-analysis of different studies. The authors have addressed this in their limitation section but I would be more happy if they elaborated a bit more about it in their discussion. I do feel this is an interesting study and it merits publication provided some minor corrections as mentioned are made!! It is not a groundbreaking study and not the most robust analysis – not because correct methodology was not followed but due to the nature of the available data and studies. Nevertheless I have not seen something similar come out and therefore I think some evidence is better than no evidence – provided limitations are addressed/mentioned for the reader to be aware.
In conclusion, good job to the authors, I am awaiting your revised work. Try to see if sub-power analysis can be performed – it will add to the work! Kind regards and stay healthy.
Thank you very much for your positive comments. According with the reviewer comments, the majority of studies in this topic are statistically underpowered, which reduces the chance that a statistically significant finding reflects a true effect.
We assumed random effect analysis and used inverse variance method. Heterogeneity was calculated with I2and also visual inspection of the forest plot was used.
Although many outcome variables were found, only manuscripts that used the same instruments to assess an outcome variable were considered for meta-analysis. In performing the different analyses, the weight of each study in the final value was noted. In fact, the sensitivity analyses carried out for the four outcome variables analysed did not change the effect.
The sample size for three outcome variables (hospital admissions, physical status and anxiety) was adequate; in the case of quality of life the sample size was low but without heterogeneity (I2=0), as only those using the Barthel scale were considered, and the effect size was adequate.
Round 2
Reviewer 3 Report
The authors clearly improved the manuscript, therefore, in my opinion, it is now susceptible for publication